# Improvement of Colonic Immune Function with Soy Isoflavones in High-Fat Diet-Induced Obese Rats

**DOI:** 10.3390/molecules24061139

**Published:** 2019-03-22

**Authors:** Qihui Luo, Dongjing Cheng, Chao Huang, Yifan Li, Chengjie Lao, Yu Xia, Wentao Liu, Xiaoxia Gong, Danlei Hu, Bin Li, Xue He, Zhengli Chen

**Affiliations:** 1Laboratory of Animal Disease Model, College of Veterinary Medicine, Sichuan Agricultural University, Chengdu 611130, Sichuan, China; lqhbiology@163.com (Q.L.); chengdongjing@126.com (D.C.); huangchao@sicau.edu.cn (C.H.); yifanli0817@163.com (Y.L.); laocj163@163.com (C.L.); xiayu113bvs@163.com (Y.X.); liuwt1986@126.com (W.L.); 2Key Laboratory of Animal Disease and Human Health of Sichuan Province, College of Veterinary Medicine, Sichuan Agricultural University, Chengdu 611130, Sichuan, China; 3College of Veterinary Medicine, Sichuan Agricultural University, Chengdu 611130, Sichuan, China; gongxx0601@163.com (X.G.); hudlo226@163.com (D.H.); LinBin344@163.com (B.L.); HXzm159@163.com (X.H.)

**Keywords:** soy isoflavones, obesity, colon, intestinal barrier function

## Abstract

*Background*: The damage to intestinal barrier function plays an important role in the development of obesity and associated diseases. Soy isoflavones are effective natural active components for controlling obesity and reducing the level of blood lipid. Here, we explored whether these effects of soy isoflavones were associated with the intestinal barrier function. *Methods and Results*: The obese rat models were established by high fat diet feeding. Then, those obese rats were supplemented with soy isoflavones at different doses for 4 weeks. Our results showed that obesity induced the expressions of pro-inflammatory cytokines, decreased the anti-inflammatory cytokine (IL-10) expression, elevated intestinal permeability, altered gut microbiota and exacerbated oxidative damages in colon. The administration of soy isoflavones reversed these changes in obese rats, presenting as the improvement of intestinal immune function and permeability, attenuation of oxidative damage, increase in the fraction of beneficial bacteria producing short-chain fatty acids and short-chain fatty acid production, and reduction in harmful bacteria. Furthermore, soy isoflavones blocked the expressions of TLR4 and NF-κB in the colons of the obese rats. *Conclusions*: Soy isoflavones could improve obesity through the attenuation of intestinal oxidative stress, recovery of immune and mucosal barrier, as well as re-balance of intestinal gut microbiota.

## 1. Introduction

Obesity is defined as a disease related to shortened life expectancy and multiple health problems [1]. Currently, the high prevalence of obesity and its related diseases is one of the major threats to public health, and there are approximately 0.5 billion obese and 1.4 billion overweight people worldwide [2]. Preventing obesity and its associated diseases is a major challenge to human. Many researchers are developing new therapies of obesity according to new insights into the pathogenesis. Therefore, the role of intestine has gained more concerns.

Gut is considered as one of the largest immune organs, and it directly contacts with nutrients. Some studies have shown that long-term high-fat diet changed the gut microbiota, which led to elevated intestinal permeability and mucosal immune responses, contributing to the development of obesity and chronic inflammation. The increased intestinal permeability, as a result of the reduced expression of tight junction proteins, such as occludin and ZO-1, boosted the translocation of bacterial lipopolysaccharide (LPS), induced TLR4/NF-κB signaling pathway, and eventually led to systemic low-grade inflammation [3] and obesity related metabolic diseases [4]. High-fat diets also induced oxidative stress, lipid peroxidation and reduce antioxidant enzyme activities in liver, kidney and heart of obese mice, while the intestinal mucosa is the most susceptible to oxidative damage. Lots of studies have suggested that maintaining intestinal homeostasis is beneficial to inhibit obesity and its related low-grade inflammation, which may represent a novel treatment for obesity and associated diseases.

Soy isoflavones (SIFs), a type of phytoestrogens, are natural active components produced by legumes. Studies have indicated its various physiological functions, including anti-cancer, anti-inflammatory and immunoregulation. Multiple studies supported the view that SIF had a good inhibitory effect on weight gain and abdominal fat deposition induced by high fat diet [5]. Zhu et al. have shown that SIF could relieve the injury of intestinal and inflammatory response induced by endotoxin in weaned piglets [6]. It could also increase the expression of β-defensins and mucins and inhibit the expressions of p38 and TLR4 in ileum. These functions may be attributed to its estrogen-like action and strong reducibility. Although maintaining intestinal homeostasis helps to relieve obesity and its related diseases, the influence of SIF on gut microbes and immune functions remain unexplained in obese rats.

Soy isoflavones have low bioavailability in small intestine, and colonic microorganisms play a key role in the metabolism of it [7,8]. For example, daidzein can be transformed to estrogen by bacteria, the activity of the latter is higher than daidzein, and it’s easier to be absorbed by colon [9]. In addition, many studies have suggested an association between obesity induced by high fat diet and an increased risk of colon cancer [10,11,12], colitis and shortened colon [13,14]. Given these reasons, we explored the influence that soy isoflavones have on colonic mucosal barrier function, which will provide new insights to the application of soy isoflavones in therapeutic approaches to obesity and its related diseases.

## 2. Results

### 2.1. SIF Attenuated Obesity Induced by High Fat Diet

After nine-week feeding, rats in the HFD group gained more weight and higher blood lipid levels than those of the chow group’s (Figure 1a,e). Then, rats in the HFD group were orally administrated with SIF for 4 weeks and continuously fed with high fat diet. Two doses of SIF obviously reduced the body weight (Figure 1b,c), and high-dose SIF significantly reduced the concentrations of TC, HDL and LDL compared with the HFD group and chow group (Figure 1f). While no significant difference was shown in food intake among groups (Figure 1d). These data suggested that SIF could mitigate obesity in male rats, which was not due to reduction in food consumption.

### 2.2. SIF Affected the Expressions of Immune Factors in Colon of HFD-Fed Rats

Intestinal immune factors involved in the maintenance of intestinal immune tolerance and the intestinal barrier integrity [15]. Previous studies have shown that HFD-fed obese mice displayed higher levels of pro-inflammatory cytokines in hepatic and adipose tissues [16]. Therefore, we detected whether SIF can influence the expressions of intestinal immune factors. Our results showed that the expressions of TNF-a, IL-4, IL-6, IL-17 and IL-18 mRNAs were elevated in the colon of HFD-fed rats, while IL-10 mRNA level was reduced. However, supplementation with SIF led to significant inhibitory effects on the mRNA expressions of TNF-a, IL-4, IL-6, IL-17 and an increase in IL-10. High-dose SIF significantly reduced IL-18 mRNA expression (Figure 2a). Similar to the gene expression, there have an increased protein level of TNF-a and a reduced that of IL-10 in the HFD group. SIF suppressed the protein level of TNF-a and the high-dose SIF increased that of IL-10 in colon compared with the HFD group (Figure 2b,c). There was no statistical difference on the level of sIgA in colon lysates among groups (Appendix A). These results indicated that SIF improved immune function in HFD-fed rats by reducing the expressions of pro-inflammatory cytokines and increasing that of anti-inflammatory cytokine in colon.

### 2.3. SIF Improved Oxidative Stress in Colon of HFD-Fed Rats

High-fat diet can induce oxidative stress and lipid peroxidation, and reduce antioxidant enzyme activities. The intestinal mucosa is the most susceptible to oxidative damage. GSH, SOD, GSH-Px and CAT are important scavengers of ROS. If the activities of T-AOC, GSH, SOD, GSH-Px and CAT decrease, the number of free radicals would exaggeratedly increase, which is harmful to lipids, proteins and nucleic acids. We found that supplementation with SIF, especially at high doses, had significant upregulated effects on the activity of T-AOC, GSH-PX, SOD and CAT (Figure 3a–e). MDA is a product of lipid peroxidation, 8-OHDG is a marker of DNA peroxidation, and protein carbonyl is an indicator of protein oxidative damage. There were higher levels of MDA and protein carbonyl in colon of the HFD group. But the administration with SIF significantly reduced levels of MDA and protein carbonyl (Figure 3f,g). However, there was no difference about the content of 8-OHDG among groups (Appendix A). Our results have shown that supplementation with SIF has the potential to improve oxidative stress in the colon of HFD-fed rats.

### 2.4. SIF Improved Intestinal Mucosal Barrier Function in Colon of HFD-Fed Rats

Given that intestinal dysbiosis in HFD-fed animals may affect gut permeability and subsequently lead to bacterial LPS going into the circulation [17] and the development of obesity and related diseases. Here, we detected the effects of SIF on tight junction proteins (occludin and ZO-1), Muc-2 expressions and the level of serum LPS. Muc-2, secreted by goblet cells, is also critical for intestinal barrier function. These tight junction proteins and Muc-2 expressions were reduced in the colon of the HFD group. Supplementation with SIF markedly increased the expression of ZO-1and high-dose SIF significantly increased that of occludin and Muc-2 (Figure 4b). Similar to the gene expression, there have a reduced protein level of occludin in the HFD group, while high-dose SIF increased that in colon (Figure 4c,d). In addition, rats in the HFD group exhibited increased concentration of LPS in serum, and the concentration of LPS was reduced after SIF consumption (Figure 4a). This suggested that supplementation with SIF has the potential to restore the expression of tight junction proteins and improve metabolic endotoxaemia induced by high fat diet. Notably, the effect of high dose SIF is most obvious.

### 2.5. Response of the Gut Microbial Structure to SIF in HFD-Fed Rats

The change of intestinal immune function, the expression of tight junction proteins and serum LPS levels are associated with specific gut microbiota composition, so we did a gut microbiota analysis using Illumina HiSeq pyrosequencing. Similar to previous studies [18,19], we found that HFD-fed affects the gut microbial community, but there was no difference among the three HFD-fed groups according to PCA analysis (Figure 5a). And there were some differences about the relative abundance of bacteria. At the phylum level, the relative levels of Fusobacteria and Actinobacteria were much lower in the HFD group, while the relative level of Firmicutes was notably higher. There were significant higher relative level of Bacteroidetes and Proteobacteria and lower ratio of Firmicutes/Bacteroidetes after supplementation with SIF (Figure 5b and Appendix A). At the genus level, there were higher relative levels of Coprococcus_1, Morganella, Lactobacillus, Ruminococcus_1, Dorea, [Eubacterium]_ruminantium group, Pasteurella and Roseburia in the HFD group. After SIF addition, the relative levels of Coprococcus_1, Morganella, Lactobacillus, Oscillibacter, Ruminococcaceae_NK4A214, Dorea, Pasteurella, Ruminiclostridium_9 and Blautia were decreased, while Faecalibacterium, [Eubacterium]_oxidoreducens group, Ruminococcaceae UCG-005, Phascolarctobacterium, Prevotella_9, Lachnospira and Bacteroides increased. In addition, high-dose SIF reduced the relative levels of [Eubacterium]_ruminantium group, Candidatus_Saccharimonas and Ruminiclostridium_9 (Figure 5c and Appendix A). 

SCFAs improve inflammation both in intestinal and extra-intestinal environments via recruiting leukocyte and producing chemokines. And the anti-inflammatory effects of SCFAs have been well characterized in epithelial and immune cell [20]. Here, we observed significant lower levels of acetic acid and butyric acid in the HFD group. SIF increased the level of butyric acid and high-dose SIF increased the level of acetic acid compared with the HFD group, while there was no difference in the level of propionic acid among groups (Figure 6).

### 2.6. SIF Regulated TLR4/NF-kB Signaling Pathway Related Genes Expression

The activation of TLRs plays a pivotal role in the inflammatory response [21]. The reduced expression levels of TLR2, TLR4 and TLR9 mRNAs or proteins can enhance protection against T cell-mediated hepatitis in mice [22], and the reduced expressions of TLR4, MyD88 and NF-κB may mediate the protective effects against LPS through reducing the level of pro-inflammatory cytokines [23]. We found that HFD-fed enriched the mRNA expressions of TLR2, TLR4, MyD88 and NF-κB p65 in the HFD group. SIF significantly reduced the mRNA expressions of TLR2 and TLR4, and high-dose SIF reduced that of NF-κBp65 (Figure 7a). Similar to the gene expression, HFD-fed increased the protein levels of TLR4 and NF-κBp65 in the HFD group, and supplementation with SIF suppressed the protein levels of NF-κBp65 and TLR4 in colon (Figure 7b,c). These results suggested that SIF may mediate anti-inflammatory and antioxidant effects by reducing TLR4 and NF-κB expressions in colon.

## 3. Discussion

Chronic inflammation is a feature of obesity. Previous researches have shown that HFD-fed obese mice were accompanied with higher levels of pro-inflammatory cytokines in hepatic and adipose tissues, including TNF-a [24], IL-1β, IL-6 [25] and plasminogen activator inhibitor-1 (PAI-1). In contrast, the level of IL-10 decreased in obese animals [16,26]. Vitale et al. [27] have shown that normal weight children who attended dietary recommendations and practiced PA exhibited a reduction in BMI, an increased level of IL-10 and a reduced level of IL-17 in salivary. The same results were found in the colon of HFD-fed rats, where we observed intestinal inflammatory cytokines TNF-a, IL-4, IL-6, IL-17 and IL-18 decreased, while IL-10 that maintain immune tolerance increased after supplementation with SIF. Several potential mechanisms may contribute to this. For example, HFD-fed could reduce expressions of ZO-1, occludin and Muc-2, alter their distribution and increase intestinal permeability [4,28,29]. When the intestinal integrity was impaired, LPS, bacteria and other metabolites produced by the pathogenic intestinal flora went through intestinal epithelial, activated the immune cells with the expressions of CD14, TLR4 and NF-κB [13,29,30,31], eventually induced abnormal expressions of cytokines [32]. Not surprisingly, a much higher level of serum LPS was observed in the HFD group. Supplementation with SIF significantly reduced the level of LPS compared with the HFD group. SIF also improved intestinal barrier function through increasing the expressions of ZO-1, occludin and Muc-2. Besides, SIF could blunt TLR4 and NF-κB expressions and alter intestinal microbiota. These might be the mechanisms for SIF to improve the expressions of immune factors and protect the intestinal barrier integrity in the colon of HFD-fed rats.

SIF improved oxidative damages in colon of HFD-fed rats. We observed that HFD-fed exacerbated the lipid, protein oxidative damages and reduced the activity of antioxidases, which is similar to the results from previous studies [33,34]. SIF, especially at a high dose, increased the enzyme activities of SOD, GSH-Px, CAT, T-AOC and improved the lipid and protein oxidative damages in HFD-fed rats. MDA is a product of lipid peroxidation, and protein carbonyl is an indicator of protein oxidative damages. Lipid peroxidation can seriously damage the cell membrane, lipoprotein, and other lipid-containing structures, such as changing the membrane fluidity and permeability, damaging DNA and proteins, and thus affect the normal cell function [35]. SOD, GSH-Px, CAT, and T-AOC are important scavengers of reactive oxygen species (ROS). The increased production of ROS or intracellular redox status are involved in the activation of NF-κB; particularly, H_2_O_2_ has been found to activate NF-kB and antioxidants have been demonstrated to block NF-κB activation [36]. In our study, SIF led to a greater reduction in the expression of NF-κB. Therefore, we speculate that SIF has the potential to improve oxidative damages, which may help to improve intestinal barrier integrity and inflammation through reducing the expression of NF-κB.

SIF can also regulate gut microbiota structure, which may explain why SIF improved intestinal barrier function and obesity. We found that HFD-fed remarkably regulated gut microbiota structure, but SIF didn’t alter gut microbiota structure in HFD-fed rats. At the phylum level, the proportion of Fusobacteria and Actinobacteria notably reduced, while the proportion of Firmicutes was remarkably elevated in the HFD group. Supplementation with SIF significantly reduced the proportion of Firmicutes and Firmicutes/Bacteroidetes ratio, increased the proportion of Bacteroidetes and Proteobacteria. Some researchers believed that the host adiposity was associated with the increased ratio of Firmicutes-to-Bacteroidetes (F/B), because Firmicutes and Bacteroidetes were related to the metabolism of carbohydrates, lipids and amino acids. HFD-fed also increased the relative abundance of Firmicutes and reduced that of Bacteroidetes and Proteobacteria in human and mice [37].

Coprococcus_1 can reduce the production of propionic acid [38]. Previous studies have found that HFD-fed increased the proportion of Oscillibacter, which was related to the damaged integrity of intestinal barrier [39,40]. Xu [40] and Zhang [41] believed that berberine improved the integrity of intestinal barrier through increasing the proportion of Phascolarctobacterium and decreasing that of Oscillibacter. Morganella and Pasteurella can induce inflammation [42,43]. Faecalibacterium can influence the production of mucin in goblet cells, maintain the proportion of various cells and intestinal homeostasis [44]. The increased number of Ruminiclostridium_9 was believed to aggravate the impair induced by methamphetamine in methamphetamine-induced conditioned place preference [45].

Here, HFD-fed increased the relative abundances of Morganella and Pasteurella and reduced that of Phascolarctobacterium. SIF reduced the relative abundances of Oscillibacter, Pasteurella, Morganella, Ruminiclostridium_9 and increased those of Faecalibacterium and Phascolarctobacterium in HFD-fed rats, which may help to reduce the colonic permeability and maintain intestinal homeostasis.

Eubacterium group [46,47,48], [Eubacterium] ruminantium [49], Eubacterium oxidoreducens [50], Ruminococcaceae. NK4A214 group and Ruminococcaceae UCG-005 [51,52,53] can increase the amount of SCFAs via SCFAs synthesis using plant polysaccharide as a substrate in cecal. Previous research indicated that quercetin and resveratrol increased the relative abundance of Ruminococcaceae_UCG-005 in obese rats [51]. A higher proportion of Ruminococcaceae_UCG-005 also may contribute to relieve inflammation and weight gain [54]. According to our data, HFD-fed reduced the relative abundance of Eubacterium oxidoreducens, and supplementation with SIF increased the relative abundances of Eubacterium oxidoreducens and Ruminococcaceae_UCG-005. Therefore, the decreased level of SCFAs may be correlated with lower abundances of Eubacterium oxidoreducens and Ruminococcaceae_UCG-005 in the HFD group, while the increased levels of acetic acid and butyric acid were related to higher levels of Eubacterium oxidoreducens, [Eubacterium] ruminantium and Ruminococcaceae_UCG-005 after SIF addition.

Previous studies supported that Prevotella [55,56], Roseburia [57] and Bacteroides [58] could synthetize SCFAs using carbohydrate as substrate, and HFD-fed significantly reduced the proportion of these bacteria. A lower abundance of them indicated the disorder of intestinal barrier function. Lachnospira, an acetate-producing bacteria [54,59], is associated with an increased level of blood glucose [60]. We found that high-dose SIF increased the relative abundances of Prevotella_9, Bacteroides, Roseburia and Lachnospira, which may be related to the improvement of colonic mucosal permeability and increased levels of acetic acid and butyric acid.

Short-chain fatty acids (SCFAs) are mainly produced by anaerobic microorganism through fermenting indigestible carbohydrates. Increasing studies have shown that SCFAs can protect the intestinal mucosa, provide energy for colon cells and alleviate inflammation [61,62]. Some disease, such as T2D [63], senescence [64], colorectal cancer [65], are accompanied with a lower level of SCFAs. Here, SIF increased the level of butyric acid, and high-dose SIF increased that of acetic acid, but there was no difference about propionic acid among groups. These results suggest that the improvement of mucosal immune function and permeability in colon may be due to the increased abundances of SCFAs and SCFA producing bacteria and the reduced abundance of harmful bacteria after supplementation with SIF.

In a word, intestinal mucosal oxidative stress, mucosal barrier, immune system and gut microbiota interact with each other, while the initial lesion and subsequent reaction are not fully understood [15]. Our findings showed the abnormities of intestinal oxidative stress, mucosal barrier, immune system and gut microbiota in obese rats. SIF could relieve oxidative damage, improve the barrier function defect, reduce the expressions of pro-inflammatory cytokines, and alter gut microbiota, implying pharmacological mechanisms for treating obesity and related diseases.

## 4. Materials and Methods

### 4.1. Animal Care and Maintenance

All animal work was carried out in accordance with the Animal Care and Use Committee Guidelines of Sichuan Agricultural University, China. After one week’s acclimatization, Eighty-four male Sprague Dawley (SD) rats (five weeks old) were randomly divided into the chow diet group (Table 1) (Dashuo, Chengdu, China; Chow, *n* = 16) and the high fat diet group (Table 2) (*n* = 48). These rats were fed with the indicated diets for nine weeks to induce obesity, and the body weight was measured weekly. The criterion for the obesity rats is that the body weight of the rats in the high fat diet group is 20% more than that of the chow group. The obese rats were further randomly divided into three groups (*n* = 16/group). They were gavaged with different doses of soy isoflavones (soy isoflavone extracts, North China Pharmaceutical Co., Ltd., Shijiazhuang, China) as described in Table 3, and continually fed with the high fat diet for four weeks. The compounds of soy isoflavone extracts, as quantified by HPLC, are shown in Table 4.

### 4.2. Body Weight, Food Intake and Sample Collection

Body weight was measured weekly, and food intake was measured every two days. A blood sample was collected from the lateral tail vein. A 1- to 2-mm section was cut in the tip of the tail with a sterile scalpel blade. Blood was then milked from the base of the tail to the tip until a sufficient volume of blood was collected. These were collected for blood biochemical analysis (Beckman CX4, Indianapolis, IN, USA) at ninth and thirteenth week. At the end of thirteenth week, all experimental animals were killed. Anesthesia was performed by intraperitoneal injection with 10% Chloral hydrate (0.4 mL/100 g). Colon tissue and cecal contents were collected under sterile conditions. Samples were preserved in liquid nitrogen.

### 4.3. Quantitative Realtime PCR

Total RNA was extracted from the colon using RNAiso Plus (TaKaRa, Dalian, China). Total RNA was subjected to reverse transcription using PrimeScript RT reagent Kit With gDNA Eraser (Perfect Real Time) (TaKaRa). Quantitative real-time PCR was performed using the Bio-Rad^®^ CFX96 PCR System (Bio-Rad, CA, USA), and the relative gene expression was normalized to internal control as β-actin. Primer sequences of the target genes are described in Table 5.

### 4.4. Western Blotting

Standard WB procedures were carried out with those antibodies: rabbit anti-TNF alpha polyclonal antibody (bs-2081R, bioss, Beijing, China), IL10 antibody (GTX632359, Gene Tex, CA, USA), rabbit anti-occludin polyclonal antibody (bioss, bs-10011R), anti-NF-κB (CST, mAb #8242, Danvers, MA, USA), TLR4 antibody (NB100-56566SS, Novus, CO, USA), mouse anti-β-actin (BM0627, boster, Wuhan, China).

### 4.5. Enzyme-Linked Immunoassay (ELISA)

The levels of blood lipopolysaccharide (LPS), 8-OHdG and sIgA in colon lysates were measured by ELISA assay kits (Shanghai Enzyme Linked Organisms, Shanghai, China) according to the manufacturer’s instructions, respectively.

### 4.6. Biochemical Reaction

The levels of GSH, MDA, SOD, GSH-Px, T-AOC, CAT were determined by biochemical kit (Nanjing Built Biology, Nanjing, China) according to the manufacturer’s instructions, respectively.

### 4.7. Gut Microbiota Analysis

Cecum content samples were snap-frozen in liquid nitrogen. DNA was extracted using the E.Z.N.A.TM stool DNA kit (OMEGA, CT, USA). Then, bacterial 16S rRNA gene V3-V4 region of each sample was amplified. Pyrosequencing was performed in the Illumina HiSeq platform. The raw data and sequencing sample information have been submitted to the SILVA database to classify.

### 4.8. Statistical Analysis

Data represent the mean and standard deviation (SD). Unpaired two-tailed student’s *t*-test, one-way ANOVA and Mann-Whitney U test were performed for all statistical significance analysis using SPSS version 17.0. * *p* < 0.05, ** *p* < 0.005; ^#^
*p* < 0.05, ^##^
*p* < 0.005.

## Figures and Tables

**Figure 1 molecules-24-01139-f001:**
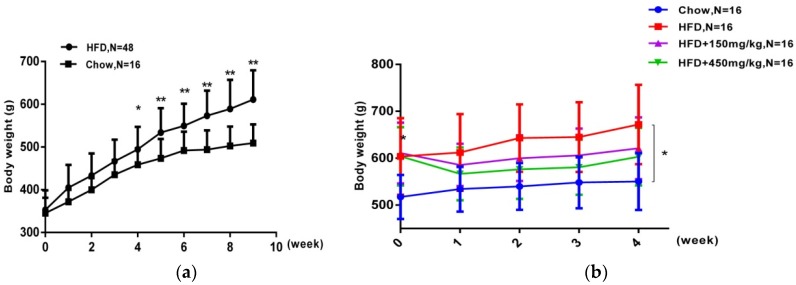
Soy isoflavones reduced the body weight and the blood lipid concentrations. (**a**) Body weights of rats in the chow group and the HFD group during nine-week fed with high fat diet; (**b**) Body weights of rats in each group during supplementation with SIF; (**c**) Body weights gain of rats in each group between week nine and week thirteen; (**d**) Food intake of a rat for one day in each group during supplementation with SIF; (**e**) The plasma TG, TC, HDL and LDL concentrations of rats in the chow group and the HFD group at the ninth week; (**f**) The plasma TG, TC, HDL and LDL concentrations of rats in each group at the thirteenth week. Error bars indicate SD. * *p* < 0.05, ** *p* < 0.005.

**Figure 2 molecules-24-01139-f002:**
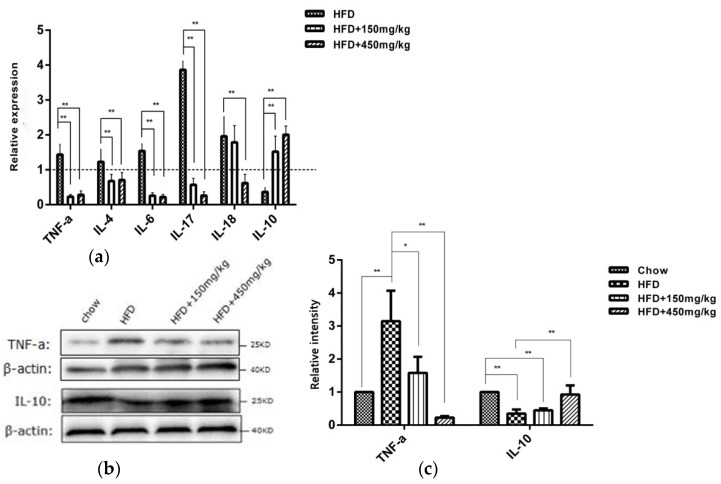
SIF affected mRNA expressions of cytokines in colon. (**a**) mRNA relative expressions (fold of chow) of TNF-a, IL-4, IL-6, IL-17, IL-18 and IL-10 in colon were assessed with qRT–PCR. (**b**,**c**) Western blots and quantification showed the protein levels of TNF-a and IL-10 in colon. Error bars indicate SD. * *p* < 0.05, ** *p* < 0.005. The dashed is the chow group.

**Figure 3 molecules-24-01139-f003:**
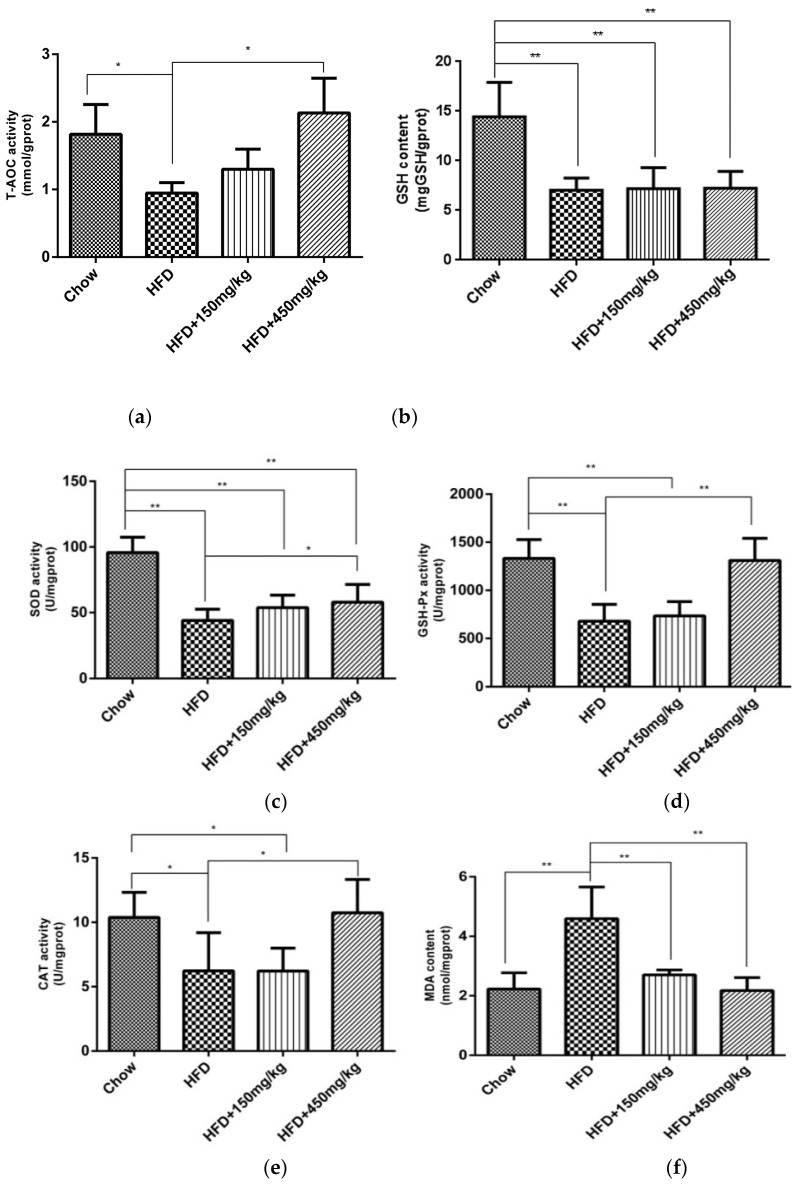
SIF improved oxidative stress and increased the activity of antioxidases. (**a**) The activity of T-AOC, (**b**) GSH, (**c**) SOD, (**d**) GSH-Px, (**e**) CAT; (**f**) The content of MDA, (**g**) Protein carbonyl. Error bars indicate SD. * *p* < 0.05, ** *p* < 0.005.

**Figure 4 molecules-24-01139-f004:**
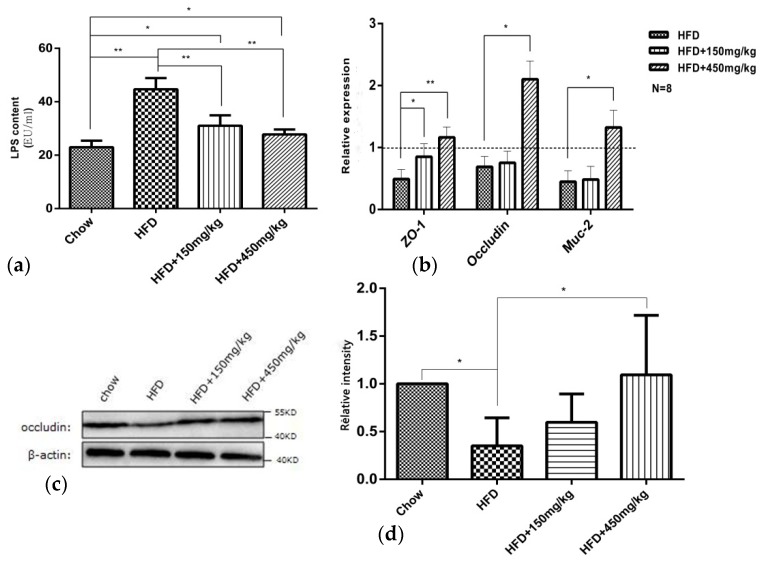
SIF protected intestinal mucosal integrity. (**a**) The plasma LPS concentrations was measured with the ELISA kit; (**b**) mRNA relative expressions(fold of chow) of ZO-1, occludin and Muc-2 in colon were assessed using qRT-PCR; (**c**,**d**) Western blots and quantification shown the protein level of occludin in colon. Error bars indicate SD. * *p* < 0.05, ** *p* < 0.005. The dashed is the chow group.

**Figure 5 molecules-24-01139-f005:**
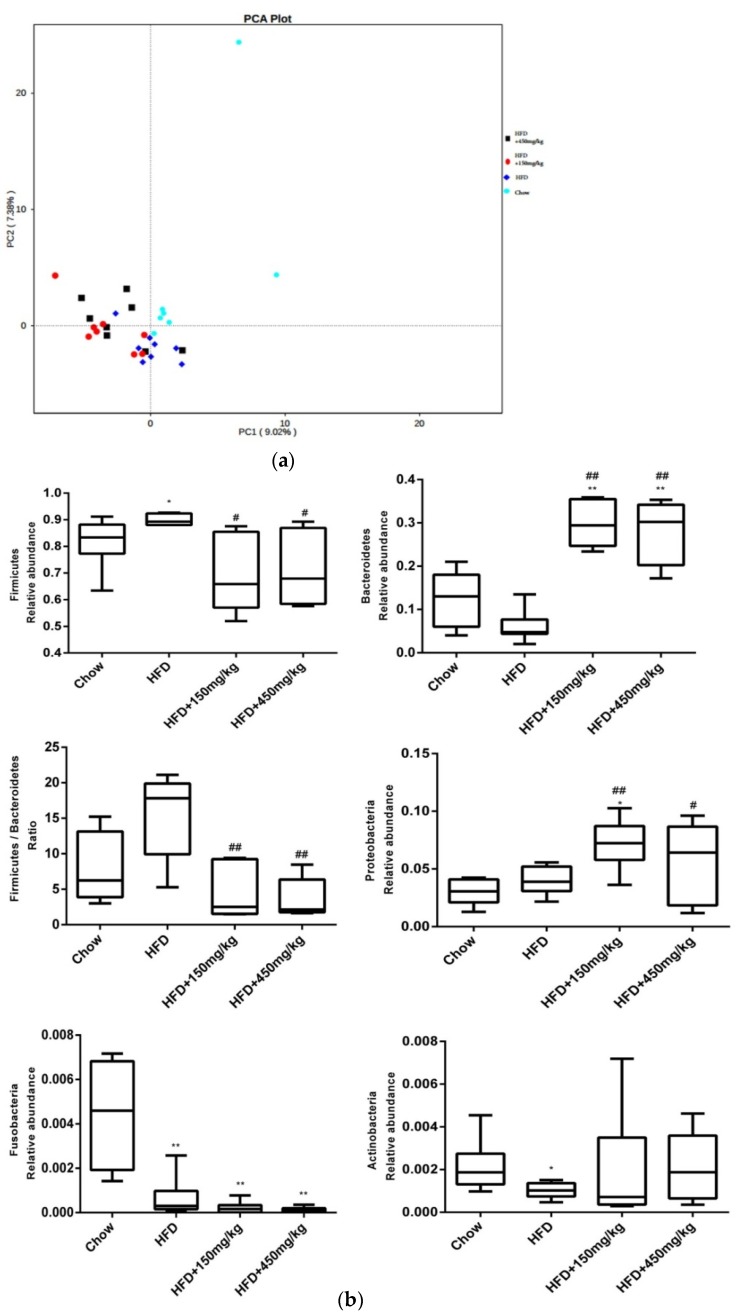
SIF altered microbiota composition in HFD-fed rats. (**a**) PCA score plot; (**b**) Relative abundance of several bacterial in the phylum (those with significant differences among groups); (**c**) Hierarchical clustering with a heat map shows the relative abundance of microbiota in the genus. *P* values were indicated as: * *p* < 0.05, ** *p* < 0.005 (* indicate versus chow group). ^#^
*p* < 0.05, ^##^
*p* < 0.005 (^#^ indicate versus HFD group).

**Figure 6 molecules-24-01139-f006:**
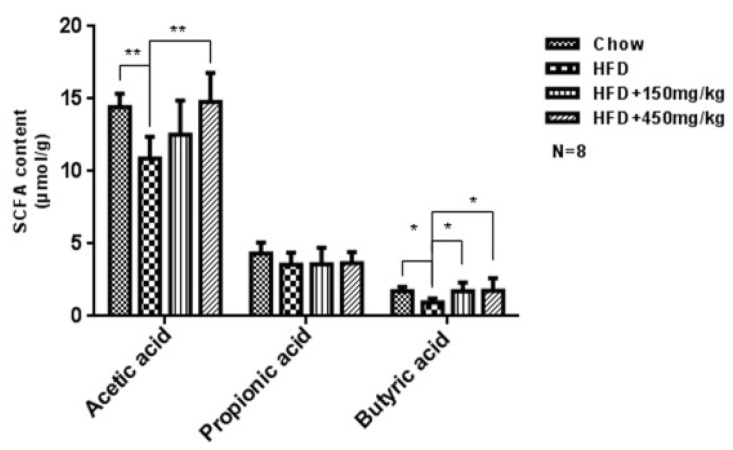
The levels of SCFAs were measured via GC on cecal fecal samples. Error bars indicate SD. * *p* < 0.05, ** *p* < 0.005.

**Figure 7 molecules-24-01139-f007:**
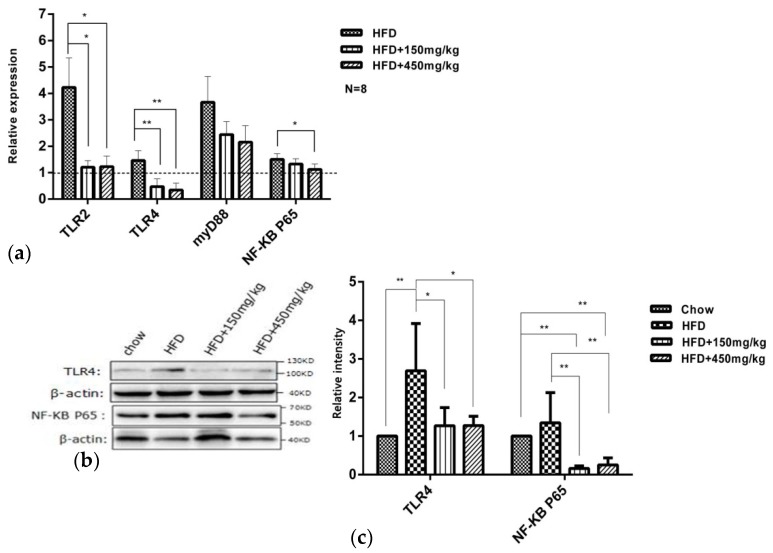
SIF modulated TLR4/NF-κB signaling pathway related genes expression. (**a**) mRNA relative expressions(fold of chow) of TLR2, TLR4, myD88 and NF-κB P65 in colon were assessed using qRT–PCR; (**b**,**c**) western blots and quantification shown the protein level of TLR4 and NF-κB p65 in colon. Error bars indicate SD. * *p* < 0.05, ** *p* < 0.005. The dashed is the chow group.

**Table 1 molecules-24-01139-t001:** Composition of the chow diets.

Ingredients	Content
Corn	54.0%
Fish meal	6.0%
Wheat bran	14.0%
Alfalfa meal	13.0%
Cotton meal	10.0%
Limestone	1.00%
Dicalcium phosphate	0.2%
Dodium chloride	0.3%
Vitamin & mineral	1.5%

**Table 2 molecules-24-01139-t002:** Composition of the high fat diets.

Ingredients	Content
Chow diets	69.5%
Pork fat	15%
Sucrose	15%
Pig bile	0.5%

**Table 3 molecules-24-01139-t003:** Composition of the experimental diets from week nine to week thirteen.

Groups	Chow Group (*n* = 16)	HFD Group (*n* = 16)	HFD + 450 mg/kg Group (*n* = 16)	HFD + 150 mg/kg Group (*n* = 16)
Diets	chow diet	high fat diet	High-dose SIF (high fat diet + 450 mg/kg SIF)	Low-dose SIF (high fat diet + 150 mg/kg SIF)

**Table 4 molecules-24-01139-t004:** Composition of the soy isoflavones extracts.

Compounds	Content
Daidzin	50.98%
Glycitin	30.36%
Genistein	8.80%
Daidzein	1.24%
Genistin	0.06%
Total isoflavones (HPLC)	91.64%

**Table 5 molecules-24-01139-t005:** Primer sequences used for the real-time PCR analysis.

Gene	Primers
Occludin	Fr 5′-AGTACATGGCTGCTGATG -3′ Rv 5′-CCCACCATCCTCTTGATGTGT -3′
ZO-1	Fr 5′- AACCCGAAACTGATGCTATGGA-3′ Rv 5′- GCGGCCTTGGAATGTATGTG-3′
MUC-2	Fr 5′-CACTGCGATGCCAACGACA -3′ Rv 5′-GCCACTAACTGCTTGTTCACCTGTA -3′
IL-10	Fr 5′-CCAGTCAGCCAGACCCACAT -3′ Rv 5′-CAACCCAAGTAACCCTTAAAGTCC -3′
TNF-a	Fr 5′- TCGTAGCAAACCACCAAGCAG-3′ Rv 5′-CAGCCTTGTCCCTTGAAGAGAA -3′
IL-6	Fr 5′- GTTGCCTTCTTGGGACTGATGT-3′ Rv 5′-TCTGTTGTGGGTGGTATCCTCTG -3′
IL-17	Fr 5′-CTGTTGCTGCTACTGAACCTGG-3′ Rv 5′-CGCTTTTGAGCTAAGGGAGTTG-3′
IL-4	Fr 5′-CGTGATGTACCTCCGTGCTTG-3′ Rv 5′-GAAGTCTTTCAGTGTTGTGAGCGT-3′
IL-18	Fr 5′-ACCTGAAGATAATGGAGACTTGGAA -3′ Rv 5′-TCTGGGATTCGTTGGCTGTT -3′
TLR 4	Fr 5′-CATTGCTGCCAACATCATCCA -3′ Rv 5′- CCAGAGCGGCTACTCAGAAACT-3′
TLR 2	Fr 5′- GAGGGAGCTAGGTAAAGTAGAAACG-3′ Rv 5′- GAGAAAGAGCAGGGAACCAGAA-3′
MyD 88	Fr 5′-TCCAGGTGTCCAACAGAAGCG -3′ Rv 5′-TGGCAAGACGGGTCCAGAAC -3′
P65	Fr 5′-ACCTGGAGCAAGCCATTAGCC -3′ Rv 5′-CGCACTGTCACCTGGAAGCA -3′
β-Actin	Fr 5′-ACGGTCAGGTCATCACTATCG-3′ Rv 5′-GGCATAGAGGTCTTTACGGATG -3′

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
