# Peer review of "Improvement of Colonic Immune Function with Soy Isoflavones in High-Fat Diet-Induced Obese Rats"

_molecules, 2019, doi:10.3390/molecules24061139_

Reviewer 1 Report

This manuscript is a well-written paper on an interesting aspect of colonic immune function by soy isoflavone. There are some comments and suggestions that the author should consider.

L29-30 Which may be the mechanisms that soy isoflavones prevent and treat obesity and its associated diseases.

Please state clearly what the authors would like to describe.

Fig.1 (e)and (f)

  The unit (m/mol) of serum concentration is not correct.

There is a remarkable difference between food intakes in the HFD and HFD+isoflavone groups. Since isoflavone is bitter, is not eating disorder caused? The difference of food intake may affect to intestinal barrier. Please consider this point.

Animal care and maintenance

Fill in the age of the rats used in the experiment.

Describe the blood and organ collection methods in detail.

Add a table of composition of high fat diet.

Generally, the amount of glycitin is less than the sum of genistin and genistein in soybean. Reanalysis of soy isoflavone extract is necessary (Table 4).

Author Response

Responses to Reviewer Comments

Point 1: This manuscript is a well-written paper on an interesting aspect of colonic immune function by soy isoflavone. There are some comments and suggestions that the author should consider.

Response 1: We appreciate the reviewer’s positive comments on our manuscript.

Point 2: L29-30 Which may be the mechanisms that soy isoflavones prevent and treat obesity and its associated diseases. Please state clearly what the authors would like to describe.

Response 2: We appreciate the reviewer’s concern, and apologize that we didn’t state this clearly. In fact, we want to say that soy isoflavones could relieve oxidative damage, improve the barrier function defect, reduce the expressions of pro-inflammatory cytokines, and alter re-balance intestinal microbiota in HFD-induced obesity. And, we have rewritten this in the revised manuscript.

Point 3: Fig.1 (e) and (f). The unit (m/mol) of serum concentration is not correct.

Response 3: We apologize for this mistake, and we have revised this.

Point 4: There is a remarkable difference between food intakes in the HFD and HFD+isoflavone groups. Since isoflavone is bitter, is not eating disorder caused? The difference of food intake may affect to intestinal barrier. Please consider this point.

Response 4: We appreciate the reviewer’s concern. These rats were given with different doses of soy isoflavones through intragastric administration. So, stress response maybe evoked resulting in difference of food intakes. However, no significant difference of food intakes between the HFD and HFD+isoflavone groups were detected during the experimental period. Thus, we think the intestinal barrier may not be affected by the food intake.

Point 5: Animal care and maintenance    Fill in the age of the rats used in the experiment. Describe the blood and organ collection methods in detail. Add a table of composition of high fat diet.

Response 5: We appreciate the reviewer’s suggestions. We have filled in these information in the revised manuscript.

Point 6: Generally, the amount of glycitin is less than the sum of genistin and genistein in soybean. Reanalysis of soy isoflavone extract is necessary (Table 4).

Response 6: We appreciate the reviewer’s concern. The soy isoflavones we used were extracted from the soybean germ, so the amount of glycitin is more than the sum of genistin and genistein. If the soy isoflavones were synthetic, the amount of glycitin could less than the sum of genistin and genistein. In addition, the compositions of soy isoflavones extracts in our study were also reported in other studies[1-6].

References

 [1]  Huang C; Pang D, Luo Q, et al. Soy Isoflavones Regulate Lipid Metabolism through an AKT/mTORC1 Pathway in Diet-Induced Obesity (DIO) Male Rats. Molecules, 2016,21(5)

[2]    陈晓林. 大豆异黄酮干预肥胖大鼠睾酮合成与分泌相关因子的研究.[四川农业大学,2016

[3]    李立科, 罗启慧, 唐秀莹, . 大豆异黄酮对肥胖大鼠肠道瘦素介导 Janus激酶/信号转导及转录激活因子信号转导通路的影响. 动物营养学报, 2016(3)

[4]    李立科, 罗启慧, 黄超, . 大豆异黄酮对雄性大鼠脾脏IL-2IL-4TNF-α、INF-γ蛋白表达的影响. 浙江农业学报, 2017,29(9):1458-1464

[5]    唐伊, 谭金龙, 罗启慧, . 大豆异黄酮对大鼠骨骼肌纤维组织形态及肌收缩蛋白表达的影响. 中国细胞生物学学报, 2018,40(08):1319-1325

[6]  陈苹, 罗启慧, 黄超, . 大豆异黄酮对健康大鼠肝脏CB1RFAAHDGAT1表达的影响. 西北农林科技大学学报(自然科学版), 2018,46(05):23-29

Reviewer 2 Report

In this manuscript entitled Improvement of Colonic Immune Function with Soy 3 Isoflavones in High-fat Diet Induced Obese in Rats", the authors have evaluated, experimentally, explored whether these effects of soy isoflavones were associated with the intestinal barrier function. In technical aspects, this authors investigated obese rat models were established by high fat diet feeding. Then, those obese rats were supplemented with soy isoflavones at different doses for 4 weeks. Authors showed investigated those chemical assay and characterization on obesity induced model. It is an interesting paper, well written and structured, worth to be published in molecules. Furthermore this study could include that soy isoflavones improve intestinal immune barrier and mucosal permeability, attenuate oxidative damages and change gut microbiota. And SIF could relieve oxidative damage, improve the barrier function defect, reduce the expressions of pro-inflammatory cytokines, and alter gut microbiota, implying pharmacological mechanisms for treating obesity and related diseases. This manuscript might be acceptable for the publication in molecules.

Author Response

Response to Reviewer Comments

Point 1: In this manuscript entitled Improvement of Colonic Immune Function with Soy 3 Isoflavones in High-fat Diet Induced Obese in Rats", the authors have evaluated, experimentally, explored whether these effects of soy isoflavones were associated with the intestinal barrier function. In technical aspects, this authors investigated obese rat models were established by high fat diet feeding. Then, those obese rats were supplemented with soy isoflavones at different doses for 4 weeks. Authors showed investigated those chemical assay and characterization on obesity induced model. It is an interesting paper, well written and structured, worth to be published in molecules. Furthermore this study could include that soy isoflavones improve intestinal immune barrier and mucosal permeability, attenuate oxidative damages and change gut microbiota. And SIF could relieve oxidative damage, improve the barrier function defect, reduce the expressions of pro-inflammatory cytokines, and alter gut microbiota, implying pharmacological mechanisms for treating obesity and related diseases. This manuscript might be acceptable for the publication in molecules.

Response 1: We appreciate the positive comments from the reviewer.